# The Underlying Relationship between Keratoconus and Down Syndrome

**DOI:** 10.3390/ijms231810796

**Published:** 2022-09-16

**Authors:** Theresa Akoto, Jiemin J. Li, Amy J. Estes, Dimitrios Karamichos, Yutao Liu

**Affiliations:** 1Department of Cellular Biology & Anatomy, Augusta University, Augusta, GA 30912, USA; 2Department of Ophthalmology, Augusta University, Augusta, GA 30912, USA; 3James & Jean Culver Vision Discovery Institute, Medical College of Georgia, Augusta University, Augusta, GA 30912, USA; 4North Texas Eye Research Institute, University of North Texas Health Science Center, Fort Worth, TX 76107, USA; 5Department of Pharmaceutical Sciences, University of North Texas Health Science Center, Fort Worth, TX 76107, USA; 6Department of Pharmacology and Neuroscience, University of North Texas Health Science Center, Fort Worth, TX 76107, USA; 7Center for Biotechnology and Genomic Medicine, Medical College of Georgia, Augusta University, Augusta, GA 30912, USA

**Keywords:** Keratoconus (KC), Down syndrome (DS), Central Corneal Thickness (CCT), cornea, genetics, mouse models

## Abstract

Keratoconus (KC) is one of the most significant corneal disorders worldwide, characterized by the progressive thinning and cone-shaped protrusion of the cornea, which can lead to severe visual impairment. The prevalence of KC varies greatly by ethnic groups and geographic regions and has been observed to be higher in recent years. Although studies reveal a possible link between KC and genetics, hormonal disturbances, environmental factors, and specific comorbidities such as Down Syndrome (DS), the exact cause of KC remains unknown. The incidence of KC ranges from 0% to 71% in DS patients, implying that as the worldwide population of DS patients grows, the number of KC patients may continue to rise significantly. As a result, this review aims to shed more light on the underlying relationship between KC and DS by examining the genetics relating to the cornea, central corneal thickness (CCT), and mechanical forces on the cornea, such as vigorous eye rubbing. Furthermore, this review discusses KC diagnostic and treatment strategies that may help detect KC in DS patients, as well as the available DS mouse models that could be used in modeling KC in DS patients. In summary, this review will provide improved clinical knowledge of KC in DS patients and promote additional KC-related research in these patients to enhance their eyesight and provide suitable treatment targets.

## 1. Introduction

Keratoconus is a corneal ectatic condition and a major indication for corneal transplantation worldwide [1]. It is characterized by asymmetric, gradual thinning and the cone-shaped protrusion of the cornea, which results in refractive defects such as myopia, irregular astigmatism, light sensitivity, visual distortion, and a reduction in total visual acuity [2,3]. KC’s progression may lead to scar formation and ruptures in Descemet’s membrane that lead to corneal hydrops. KC typically presents in adolescence, progresses for 10–20 years, and stabilizes in the third or fourth decade of the individual’s life [4,5]. Early estimates indicated that the incidence of KC was around 1 in every 2000 people globally; however, newer research has estimated its frequency to be 1 in every 350 people in the general population [6,7,8]. This increase in cases ranges widely among ethnic groups [9,10,11,12,13,14,15], possibly due to the different diagnostic criteria used between studies and technological advancements [2,9,16,17]. The cause of KC is unknown; however, it is usually linked to genetics, environmental variables such as eye rubbing, atopy, ultraviolet exposure, hormones, the inflammatory response, and pregnancy [3,17,18,19,20,21]. Aside from these often-reported connections, there is a substantial volume of literature on KC comorbidities such as Ehlers–Danlos disease, Marfan syndrome, mitral valve prolapse, floppy eyelid syndrome, and Down syndrome (DS) [22,23,24,25,26,27,28,29]. 

DS is the most prevalent chromosomal anomaly in newborns and the most common cause of intellectual impairments due to chromosomal nondisjunction [30]. The trisomy of chromosome 21 leads to many dysmorphic features primarily affecting the head and neck [31]. Besides these systemic pathologies, individuals with DS have an increased risk of several ocular diseases [30,32,33,34,35]. Recently, there have been increased reports of KC risk in DS patients [28,36,37]. The published literature suggests patients with DS exhibit corneal morphologic findings indicative of KC [30,38]. Furthermore, research has revealed that DS patients have steeper and thinner corneas, which are phenotypes related to KC [39,40,41]. Additionally, a significant reduction in corneal thickness was also seen in participants with DS when compared to control healthy groups [42]. Although no clear cause–effect link has been demonstrated, the known frequent forceful eye rubbing in people with DS has been hypothesized as one of the contributing risk factors that predispose DS individuals to corneal abnormalities, particularly to the development of KC [18,36,43,44]. Therefore, this study sought to review studies of KC in DS by addressing the association of KC and DS with respect to genetics, central corneal thickness, and mechanical forces such as vigorous eye rubbing. In addition, we review KC’s diagnosis and treatment options, which may be relevant in detecting KC in DS patients. We also discuss available DS mouse models, which may be relevant in studying the KC–DS connection. Overall, this review contributes to a better clinical understanding of KC in DS patients and raises awareness on the need to improve their vision and provide relevant therapeutic targets.

## 2. Ophthalmic Manifestations of Down Syndrome

Ophthalmic manifestations are common in patients with DS, with frequently reported refractive errors, strabismus, and nystagmus [45,46]. Refractive errors typically occur in at least one eye of patients with DS, with hyperopia and astigmatism most commonly seen in Europeans, Americans, Koreans, and Brazilians [32,47,48]. Strabismus has been reported in association with DS, with esotropia generally more common than exotropia [49]. Cataracts may occur in DS newborns, with corneal opacification more likely to develop as the patients age [50,51]. Nystagmus has increased prevalence in DS patients and may cause significant visual impairment [34,52]. Other ocular defects associated with DS include retinal dystrophy, fundus depigmentation, and choroidal sclerosis [53,54]. 

Though less prevalent than some other ophthalmic characteristics, KC has been shown to have a prevalence ranging from 0% to 71% in DS patients of various ethnicities [18,28,32,55,56]. Contradictory studies suggest no link between DS and KC [47,57,58,59]. Several reasons might explain the discrepancy in DS patients and KC’s prevalence, including the clinical equipment utilized, ethnicity, the lack of cooperation from patients during screening, the diagnostic criteria for KC, sample size, and the patient’s age during clinical examinations (<18 years) [30,42,59,60]. It is well known that KC manifests itself throughout the adolescent years, although some are reported to have an onset <10 years old [5,61,62,63,64]. In a study by Alio et al., the authors noted that approximately 75% of 112 DS patients, comprising 59% Caucasians and 41% Arabs, demonstrated corneal features likely to be characteristic of KC [30]. For example, they found that patients with DS have steeper and thinner corneas and more corneal aberrations than those without any genetic alterations [30]. For these studies, the authors collected the data using the Sirius system. This specific topographical platform combines a spinning Scheimpflug camera with a Placido disk to enable a thorough study of the cornea by measuring 35,632 and 30,000 points on the anterior and posterior corneal surfaces, respectively [30]. The studies by Bermudez et al. reported a prevalence of KC of 27.2% among 1207 Brazilians with DS using slit lamp examination, corneal pachymetry, and topography [65]. The studies by Doyle et al. reported a prevalence of KC of 2% in 50 individuals with DS in Manchester area using slit-lamp examination [66]. Clearly, the prevalence of KC in DS varied between these studies possibly due to sample size, different instrumentation, and ethnicity.

In the studies by Aslan et al., age and imaging techniques also accounted for the varied observation of KC in DS patients [42]. The authors observed that 27 children with DS with a mean age of 8.84 years had thinner corneas with a mean CCT of 494.27 ± 47 µm compared to healthy participants with a mean CCT of 539.3 ± 40 µm, suggesting a higher risk of KC in these children [42]. Using Scheimpflug imaging, the authors measured both central and ectatic corneal thicknesses allowing for the early detection of KC in this DS cohort [42]. The studies by van Splunder et al. did not find any significant correlation between KC and DS among 1539 adults in the Netherlands. The lack of correlation might be due to the advanced age (≥50 years) of the DS patients in the study [55]. Similar observations were delineated by Wong et al., who did not detect KC in DS patients due to the very young age of the children, with a mean age of 3.74 years, which is presumably too young for KC to be detected clinically [58]. Table 1 summarizes the literature reports after a PubMed literature search with the phrase “Keratoconus and Down syndrome” for studies of KC in DS in both young and adult patients from different ethnic backgrounds. After careful screening, nineteen studies were selected based on all the available information on the sample size, mean age/age range of DS patients, ethnicity, and diagnosis inclusion criteria, as these parameters are all essential in connecting KC with DS. A study was not selected if any of these parameters were missing. Out of the nineteen, ten studies reported the incidence of KC in DS, four studies reported patients with DS having steeper and thinner corneas relative to control groups, and five studies observed no association between KC and DS. Not surprisingly, KC was observed in DS patients ranging from a mean age of 11.7 years to 48.46 years, reinforcing the occurrence of KC in teenage years, progression in mid-20’s, and stabilization in the third of fourth decade of an individual’s life [67]. 

## 3. Genetics of Keratoconus and Down Syndrome 

Numerous genetic studies have suggested associations between KC and DS [18,73,74]. A recent genome-wide association study (GWAS) aiming to discover novel genomic loci associated with KC identified rs7674734 (*p* = 1.34 × 10−13) in chromosome 21 (Chr21q2) associated with KC, suggesting a potential genetic link with DS [36,75]. Vincent et al. indicated that a genetic relationship between KC and DS might be due to the dosage impact of trisomy 21, which can affect a gene locus for KC on chromosome 21 or influence another gene on another locus that may be associated with KC [41]. Furthermore, the phenotypic changes in DS may be attributed to a “gene-dosage effect”, where the extra copy of the gene on chromosome 21 in DS patients might lead to the overexpression of the phenotypic changes usually associated with the normal genes [76,77].

Central corneal thickness is an important indication of corneal health [78]. One of the significant risk factors for KC in people is a decreased CCT [75]. A number of chromosomal regions comprising the *COL1A1*, *COL1A2*, *COL4A1*, *COL4A2*, and *COL5A1* genes have been associated with CCT variations and KC risk in various ethnic groups [79,80]. The corneal stroma accounts for approximately 90% of the total thickness of the human cornea and is mostly made up of collagen [81]. These collagen proteins are decreased in KC, leading to the weakening of the corneal stroma and the subsequent reduction in their mechanical stability [82,83,84,85,86]. As a result, the incidence of KC is thought to be connected with the dynamics in collagens of the corneal stroma [83,87]. In a genome-wide association study, significant associations with KC have been identified within or close to genes including *COL1A1* (rs2075556, *p* = 3.35 × 10−09), *COL5A1* (rs3118518, *p* = 1.83 × 10−28), *COL6A1* (rs142493024, *p* = 9.07 × 10−12), and *COL12A1* (rs35523808, *p* = 2.90 × 10−25) [75]. 

Collagen types I and V are the most prevalent in the stromal layer of the human cornea, with comparatively lower levels of collagen types VI, XII, XIII, and XIV. All these collagen proteins are encoded by separate genes located on various chromosomal loci [83,88]. Although present in small proportions, collagen VI may have an integral function in the development of KC and DS [89,90,91,92,93]. Collagen VI is encoded by six distinct genes (*COL6A1*, *COL6A2*, *COL6A3*, *COL6A4*, *COL6A5*, and *COL6A6*) that code for proteins that contribute to the formation of numerous tissues, including the human cornea [94,95]. Unsurprisingly, Kabza et al. discovered a substantial reduction in *COL6A1* and *COL6A2* in KC-affected human corneas after performing a whole transcriptome profile of human KC corneas using an RNA-sequencing technique [96]. *COL6A2* was also discovered as one of the differentially expressed genes by Hao et al. utilizing human corneal fibroblast cells from probands with KC and healthy individuals [97]. The altered expression of *COL6A2* may impact the expression of related collagens or extracellular matrix (ECM) proteins, leading to lower ECM levels in corneas and KC development [97]. Interestingly, several studies have implicated the functional involvement of COL6A1 and COL6A2 in the development of different phenotypes among DS patients. Gittenberger-De Groot et al. used immunohistochemistry to confirm an overexpression of COL6A1 and COL6A2 in the hearts of trisomy 21 fetuses, suggesting their potential involvement in the development of congenital cardiac abnormalities seen in DS patients [98]. Many follow-up studies have provided additional data to further support this finding [99,100]. Reports of *COL6A1* and *COL6A2* gene expression have also been shown in nuchal skin and the umbilical cords of trisomy 21 fetuses [93,101,102]. Given the location of the *COL6A1* and *COL6A2* genes on human chromosome 21 and their role in KC and DS, these genes may underlie the pathological connection between KC and DS [2,98,103,104,105]. 

Superoxide dismutase 1 (*SOD1*) is another potential gene that has been linked to KC and that directly participates in antioxidative activities [29,106,107,108]. *SOD1* gene consists of five exons located on chromosome 21q22.11. The encoded protein SOD1 binds to molecules of copper and zinc (CuZn-SOD) to eliminate superoxide radicals in the body [109]. The CuZn-SOD enzyme is present in the human cornea and is an essential antioxidant enzyme [109]. Given the substantial buildup of oxidative stress indicators in the corneas of KC patients, there is evidence of a connection between *SOD1* mutations and the development of KC [108]. Studies by Udar et al. identified a 7-base intronic deletion in the *SOD1* gene in two KC families after screening the exons and intron–exon junctions of *SOD1* using PCR-based Sanger sequencing [109]. Additionally, an mRNA analysis of one affected individual confirmed two alternatively spliced transcript isoforms (one lacking exon 2 and one that lacked both exon 2 and exon 3) coding for proteins lacking the active site of the SOD1 enzyme [109]. Although many other reports with different human cohorts have not identified *SOD1* mutations in KC patients [110,111,112], a few studies have verified the existence of *SOD1* mutations in KC patients, indicating the probable involvement of SOD1 in the pathogenesis of KC [107,113,114]. Increased oxidative damage and the involvement of the *SOD1* gene have also been implicated in Down syndrome [115,116,117,118,119,120,121]. Hence, given the location of the *SOD1* gene and its role in increasing oxidative damage with its mutations, an association of DS with KC is evident. With all these findings, it will be interesting for researchers to further study the molecular mechanisms of the collagen VI family of genes (*COL6A1* and *COL6A2*) and the *SOD1* gene to elucidate their target genes and their functional involvement in both KC and DS.

## 4. Central Corneal Thickness (CCT) of Keratoconus and Down Syndrome Patients

The thickness of the central cornea is a critical risk factor that has been implicated in the pathophysiology of KC [122,123]. For human eyes without KC, the central and peripheral corneal thickness ranges from 551–565 µm and 612–640 µm, respectively, when measured using ultrasonic pachymetry, Galilei, and Orbscan II [122]. Several factors such as age, ethnicity, gender, time of day, the instrument used, and the rate of blinking have been reported to account for variations in CCT in humans [124,125,126]. Since KC usually presents itself during teenage years or early adulthood, CCT influenced by age could be a significant predictor of KC in DS patients. Several studies have reported that DS patients have significantly reduced corneal thicknesses at different ages [40,42,127,128].

Apical corneal thickness (ACT) in 10–20- and 21–30-year old DS patients was substantially lower with a mean ACT of 518.4 ± 33.9 µm and 511.2 ± 29.4 µm compared with non-DS individuals with a mean ACT of 556.6 ± 31.2 µm and 552.0 ± 39.7 µm, respectively [127]. In addition, Evereklioglu et al., in a case–control study, compared CCT measurements by ultrasound pachymetry in DS children with a mean age of 9.28 ± 3.47 years and age- and sex-matched healthy controls with a mean age of 8.75 ± 3.30 years. They observed that children with DS have decreased CCT with a mean of 488.39 ± 39.87 µm relative to the healthy group with a mean of CCT of 536.25 ± 20.70 µm [128]. Further, Haugen et al., in a population-based study, observed a reduced CCT of 480 ± 40 µm in DS patients versus a CCT of 550 ± 30 µm in non-DS controls [40]. Similar findings were also reported in DS children with a mean CCT of 494.2 ± 47 µm compared with healthy control groups with a mean CCT of 539.3 ± 40 µm [42]. 

Conversely, other findings have associated thicker corneas with DS, although these studies included patients with other intellectual disabilities (ID). Akinci et al. determined CCT in 77 ID individuals including those with DS [129]. From their study, 23 of these 77 ID patients had DS as the cause of their ID [129]. Their studies suggested an increase in CCT with a mean of 554.8 ± 14.5 μm in DS patients compared to their idiopathic group with a mean CCT of 539.0 ± 12.1 μm [129]. In addition, there was no significant difference in CCT between DS patients and those with other syndromic ID etiologies, with a mean CCT of 565.4 ± 30.3 μm [129].

Aside from CCT, several corneal morphological changes indicative of KC have been reported in DS patients [28,30,36,40,45,46,130]. These changes include steeper anterior and posterior corneal surfaces, increased corneal aberrations (both coma and coma-like), thinner corneas, and irregular posterior corneal surfaces. Although CCT may not be the ideal way of characterizing KC in DS patients, it may provide relevant indications of the possibility of KC coupled with the observation of other corneal morphological characteristics in DS patients. It is necessary to measure CCT and other corneal features in DS patients, especially those with KC.

## 5. Eye Rubbing in Keratoconus and Down Syndrome Patients

Several studies have suggested vigorous eye rubbing as a potential risk factor of KC in DS patients [74,131,132]. Vigorous eye rubbing in KC patients exerts pressure on the cornea resulting in the potential dysregulation of proteolytic enzymes [44,131]. The dysregulation of proteolytic enzymes such as matrix metalloproteinases (MMPs) degrades collagen, proteoglycans, and other components of the corneal stromal extracellular matrix resulting in corneal thinning in KC patients [83,133,134,135]. Elevated levels of collagenase (MMP-1), gelatinase A (MMP-2), gelatinase B (MMP-9), and collagenase-3 (MMP-13) have been implicated in the pathophysiology of KC [135,136]. The overexpression of MMP-1 in the corneas of KC patients induced by extracellular matrix metalloproteinase inducer CD147 (EMMPRIN) results in the degradation of type I and III collagens in the stromal layer [137]. The increased expression of KC-specific MMP-2, a major protease actively synthesized by the keratocytes, leads to the disruption of type IV basement membrane collagen and type 1 collagen [138,139]. MMP-9 has been shown to contribute to the degradation of types I and IV collagen fibers in the human corneal epithelial layer in response to injury or stress [135,140]. These MMPs (-1, -2, -9, -13) can be upregulated by cytokines such as interleukin 1β (IL-1β), IL-6, and tumor necrosis factor α (TNF-α), which are elevated in serum samples from DS patients [141,142,143,144]. Thus, the increased susceptibility to inflammatory processes following mechanical forces exerted on the cornea due to eye rubbing could partially contribute to the increased prevalence of KC in DS patients. We have reviewed different findings after a PubMed literature search to reinforce the claim of frequent eye rubbing as one of the underlying causes of frequent KC in DS patients (Table 2). Articles with clear reports on sample size, ethnicity, and sentences linking eye rubbing to KC were selected. These fifteen different studies cover different ethnic groups, ranging from Chinese/Japanese/Malaysians to Arabs/Hispanics/Caucasians. Only two studies included more than 100 subjects while the other 13 studies had a relatively limited sample size (1 to 91). Almost all the studies indicated the critical impact of vigorous eye rubbing on KC pathogenesis, although also reporting the difficulty of recapitulating that activity in any animal/in vitro model.

## 6. Diagnosis/Treatment of Keratoconus in Down Syndrome Patients

Due to the myriad of ocular symptoms associated with DS, diagnosing and treating KC in its early stages is necessary to prevent its progression and complications. The available clinical diagnostic tools include corneal topography, corneal tomography, the Belin Ambrosio Enhanced Ectasia Display, the Holladay 6 map display, corneal pachymetry, automated detection programs, and corneal biomechanics [87]. Early signs of KC include the scissoring of the red reflex (during retinoscopy or ophthalmoscopy), oil droplet red reflex, and Rizzuti’s sign, a conical reflection on the nasal cornea when a penlight is shone on the temporal side [152,153]. A slit-lamp examination can be used to detect moderate to advanced forms of KC with Munson’s sign (a V-shaped cornea during a downward gaze), Fleisher’s ring (iron deposits in the corneal epithelium), Vogt’s striae (fine vertical lines in the deep stroma and Descemet’s membrane), stromal scars, and an increased visibility of corneal nerves in DS patients [84,154,155,156]. 

The clinical management of KC is dependent on the stage of the condition and is geared towards either improving visual acuity using spectacles, hybrid lenses, piggyback, or scleral lenses, or halting disease progression using surgical modalities ranging from the use of implantable intracorneal ring segments, corneal collagen cross-linking, and, in the worst forms, corneal transplantation [84,157,158,159,160,161,162]. These different diagnostic and treatment modalities may identify and help treat the associated effects in DS patients. 

However, the detection and management of KC in DS patients have been challenging. For example, KC has been observed to occur in between 0% and 30% of DS children [32,33,47]. Several factors could account for this. The lack of cooperation—especially in DS children—during the corneal examination limits the possibility of the early detection of KC [30,57]. Further, the detection of KC in DS children is contingent on seeking medical attention and consent from parents/guardians for visual screening; thus, if there are no timely visits by parents/guardians, the early diagnosis and detection of KC in DS children may not be possible [28,163]. In addition, individuals with DS usually have speech difficulties and behavioral characteristics such as an attention problem; hence, they may not be able to communicate vision challenges or focus properly during examinations, making examinations stressful for physicians [164,165]. 

## 7. Mouse Models of DS for KC

Mice are used extensively in biomedical research to serve as model organisms in studying human diseases and biological processes and to explore the potency of candidate drugs used in disease treatments [166,167]. The mouse genome shares a 99% similarity with the human genome [168]. Animal models for KC have been very limited in their availability. KC-relevant phenotypes have been reported in a 15-year-old rhesus monkey [169], a strain of myopathic hamster [170], and two inbred strains of mice [171,172]. Recently, human genetic studies of a four-generation family identified potential mutations in the phosphatase domain of the *PPIP5K2* gene [173]. A mouse model with the loss of phosphatase activity and an elevated kinase activity showed abnormal corneal curvature alterations [173]. A number of mouse models have been found to have reduced CCT, mimicking a disease condition called keratoglobus with thin cornea, but not KC [174,175,176,177]. It is necessary to develop additional mouse models for KC research. Considering the significantly elevated prevalence of KC in DS patients, it will be interesting and necessary to examine the corneal phenotypes in DS mouse models for KC. 

Mouse models have been shown as great tools for fully understanding the phenotypic changes associated with DS, especially because the mouse genome can be genetically engineered to mimic the disease condition in DS patients [178]. Segments of the long arm of human chromosome 21 are conserved in chromosomal segments of three mouse chromosomes: 10, 16, and 17. A number of available mouse models with trisomic regions of these mouse chromosomes have been used to investigate the molecular mechanisms underlying different clinical phenotypes in DS patients (Figure 1) [179]. Most of these transgenic mice are available at the Jackson Laboratory Mice Repository through the Cytogenetic and Down Syndrome Models resource. To address KC in DS, mouse models of DS may serve as a good way of understanding or studying KC in DS, since few studies have focused on the ocular features in these DS mouse models. A mouse model of DS—murine trisomy 16—has been shown to have developmental and differentiation disorders of the corneal epithelial cell layers and structural disturbances of the corneal parenchyma [180,181]. These minor anomalies of the cornea could have resulted in KC if these DS mice had survived to adulthood [180]. We have summarized the clinical phenotypes and their limitations in different DS mouse models for their potential application in corneal research (Table 3). A PubMed literature search with the phrase “mouse models in DS” was used to compile the articles. The selection criteria of the articles used for the review was based on clear descriptions of the mouse models, their clinical phenotype(s), and the models’ limitation(s). The characterization of these mouse models may be relevant for identifying KC-related corneal phenotypes.

## 8. Summary and Future Directions

KC is the most common corneal ectatic disorder. It often occurs with several comorbidities, including genetic disorders such as Down syndrome. Increasing evidence supports the elevated risk (>100 times) of KC in DS patients. The genetic association of sequence variants within or near the *COL6A1* and *COL6A2* genes on Chr21 with KC provides an additional functional link between KC and DS [97,98]. Thus, it is imperative to unravel the functional connection and the related molecular mechanisms of these genes in KC and DS pathogenesis. Altered CCT, other corneal morphological characteristics such as steeper and thinner corneas, and imbalanced inflammatory proteins detected in the cornea may provide alternative clues useful for identifying KC in DS patients. Further, the available animal models of DS need to be characterized for their potential ocular abnormalities in relation to KC. The identification of KC-relevant corneal phenotypes in these DS mouse models will not only validate the connection between KC and DS but also provide potential animal models for KC research. In addition, with the myriad of sophisticated technology developed in clinics over the past decade, it may be possible to detect KC-related phenotypes through the continuous clinical screening and monitoring of DS patients. These DS patients may not have KC-relevant corneal phenotypes but develop KC at later ages. Clinical exams with novel technologies will enable this clinical follow-up of KC development in DS patients. Generally, as supported by these observations, it is vital that patients with DS be screened early for ocular abnormalities—especially during teenage years—to identify the early signs of KC in order to provide a better management of the progression of KC. In summary, we have presented an overview to support the strong connection between KC and DS with respect to genetic studies, corneal features, clinical diagnosis and phenotyping, environmental factors such as eye rubbing, and available mouse models (Figure 2). Additional research to advance our understanding of the underlying relationship between DS and KC could provide potential therapeutic targets for KC management and prevention in DS patients.

## Figures and Tables

**Figure 1 ijms-23-10796-f001:**
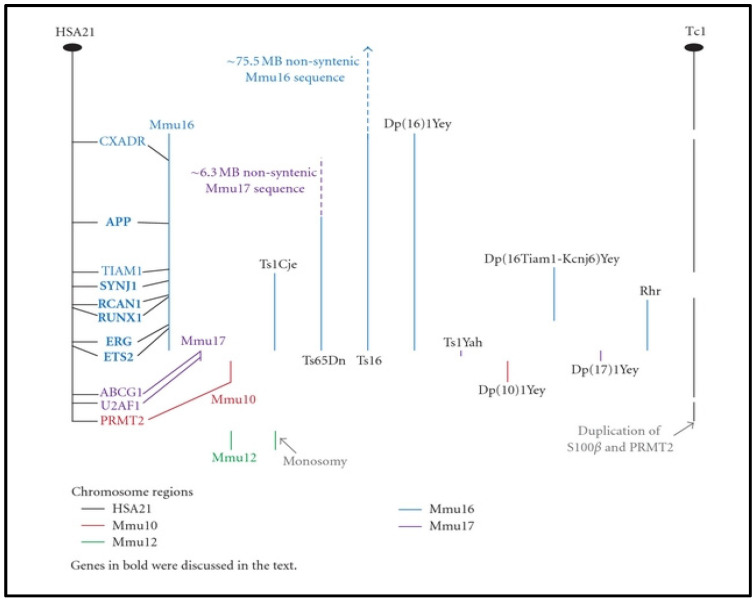
**A graphical representation of HSA21 and the syntenic mouse chromosome regions from Mmu10, Mmu16, and Mmu17.** The trisomic (or monosomic) chromosome regions present in 10 of the segmental mouse trisomies are also shown, with color-coding indicating the chromosome source of the region (see the key in the figure). The location of 11 HSA21 genes is shown, as well as their location on the syntenic chromosome regions, with text color indicating each syntenic chromosome. The location of 11 HSA21 genes is shown, as well as their location on the syntenic chromosome regions, with text color indicating each syntenic chromosome. The dark ovals indicate the HSA21 centromere (Figure 1 and legend adapted from [181]).

**Figure 2 ijms-23-10796-f002:**
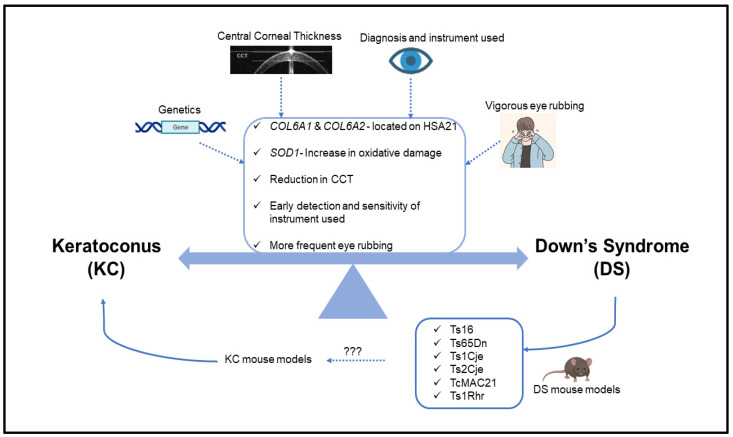
**Schematic diagram summarizing the underlying relationship between KC and DS.** Genes such as (*COL6A1*, *COL6A2*, and *SOD1*), the measurement of CCT, habitual eye rubbing, screening DS patients for KC at an early age, diagnosis criteria, and type of instrument used in detecting KC could serve as potential connections between KC and DS. In addition, screening the ocular features of DS mouse models for KC could potentially provide mouse models of KC to better understand KC in DS.

**Table 1 ijms-23-10796-t001:** Literature studying the relationship between Keratoconus and Down syndrome.

Title of Article	Sample Size	Mean Age/Age Range of DS Patients	Ethnicity	KC Diagnosis and/or Instrument Used	Findings	Reference
Corneal Morphologic Characteristics in Patients with Down Syndrome	112 DS patients	14.88 ± 15.76 years Range: 3 months–60 years	DS group—59% Caucasians, and 41% Arabs	Visual, refractive, anterior, and posterior corneal characteristics were assessed and compared in both groups using Placido disc/Scheimpflug camera topographer (Sirius, CSO)	Patients with DS have steeper, thinner corneas and more corneal aberrations compared to those without genetic alterations.	[30]
Ophthalmological abnormalities in Down syndrome among Brazilian patients	1207 DS patients	No mean age reported Range: 0–42 years	Brazilians	Slit-lamp evaluation, corneal pachymetry, and topography were used for KC detection	A prevalence of KC of 27.2 % was observed among 1207 persons with Down syndrome	[65]
Emmetropisation, axial length, and corneal topography in teenagers with Down’s syndrome	50 DS patients	17.4 years Range: 15–22 years	Mancunians	TMS-1 machine was used for corneal topographic mapping. Slit lamp examination was also conducted	Prevalence of KC in this cohort was 2%, but 6% had corneal topography with inferior steepening, which may represent a preclinical keratoconic process.	[66]
Clinical profile and main comorbidities of Spanish adults with Down syndrome	144 DS patients	35 ± 12 years Range: 17–65 years	Spanish	Not mentioned	KC is a prominent ocular feature associated with DS in individuals > 50 years.	[68]
Corneal thickness measured by Sheimpflug imaging in children with Down syndrome	27 DS patients	8.94 ± 2.35 years Range: 5–12 years	Turkish	CCT, TP, and CV were analyzed using Pentacam Scheimpflug imaging system	Corneal thickness was lower in DS children than in healthy control subjects.	[42]
Prevalence of ocular abnormalities in adults with Down syndrome in Hong Kong	91 DS patients	38 years ± 6.5 years Range: 30–56 years	Chinese	Corneal pachymetry was measured using a handheld ultrasound pachymeter	There is a high prevalence of corneal problems (including KC) in Chinese DS patients.	[43]
The ophthalmic anomalies in children with Down Syndrome in Split-Dalmatian County	153 DS patients	11.7 ± 3.2 years Range: 0–18 years	Croatians	Biomicroscopic examination of the anterior eye segment	Though less common than other refractive disorders, KC was observed in patients with DS.	[57]
Incidence of ocular pathologies in Italian children with Down Syndrome	157 DS patients	5.28 years Range: 1 month –18 years	Italians	Slit lamp biomicroscopy	No KC was observed in this cohort of DS patients.	[47]
Ocular findings in Malaysian children with Down syndrome	60 DS patients	6.72 ± 3.38 years Range: 1 month –17 years	Malaysians	Slit lamp biomicroscopy and Placido disc	No KC was observed in this cohort of DS patients.	[59]
Computerized corneal topography in a pediatric population with Down syndrome	21 DS patients	6.9 years Range: 10 months–18 years	Caucasians	Slit-lamp biomicroscopy, scissoring or oil-drop reflex on retinoscopy, corneal topography mapping using using the EyeSys Computer-Assisted Videokeratoscope	Corneal curvature in DS children was significantly steeper than in the control population.	[41]
Prevalence of ocular diagnoses found on screening 1539 adults with intellectual disabilities	1539 DS patients	45.7 years Range: 20.2–88.7 years	Dutch	Slit-lamp biomicroscopy	KC is independently related to DS (OR 7.65, 95% CI, 3.91 to 14.96).	[55]
Characteristic ocular findings in Asian children with Down syndrome	123 DS patients	6.5 years Range: 6 months–14 years	Koreans	Visual acuity assessment, slit-lamp biomicroscopy	KC was not observed in this cohort of Asian DS patients.	[33]
Biometric measurements of the eyes in teenagers and young adults with Down syndrome	47 DS patients	20.0 ± 3.9 years Range: 14–26 years	Norwegians	Retinoscopy reflexes, central corneal thickness and anterior chamber depth were measured with a A Nidek Model EAS-1000 anterior eye segment analysis system. Corneal topography was analyzed with a TechnoMed C-Scan V2.0.0. Corneal curvature was also measured with a hand-held Nidek autokeratometer KM-500	DS patients have thinner corneas and higher keratometry values compared to control.	[40]
Ocular abnormalities in Down syndrome: an analysis of 140 Chinese children	140 DS patients	3.74 years Range: 3 months–13 years	Chinese	Visual acuity assessment, slit-lamp biomicroscopy	No KC was observed in this DS cohort.	[58]
Health care concerns and guidelines for adults with Down syndrome	38 DS patients	36.2 years for middle aged DS patients Range: 30–43 years old 59.7 years for elderly DS patients Range: 47–68 years old	Canadians	Not mentioned	KC was observed in 15.8% of all DS adults.	[69]
Corneal ectasia in mothers of Down syndrome children	77 MDS and 63 MNC controls	Mean age at the time of examination was 48.81 ± 6.93 years in MDS and 48.46 ± 8.35 years in MNC	Iranians	Slit-lamp biomicroscopy, corneal tomography was assessed using Oculus Pentacam HR and corneal biomechanics was also measured using Corneal Visualization Scheimpflug Technology	Mild KC was identified in 30.9% (21 cases) in the MDS in comparison with 14.3% (9 cases) MNC.	[70]
Prevalence of Keratoconus in Persons with Down Syndrome in a National Registry in Norway	4342 DS patients	37.1 years Range: Not reported	Norwegians	Data analysis from International Classification of Diseases and Related Health Problems, Tenth Revision diagnosis codes Q90 for Down syndrome and H18.6 for KC from 1 January 2010, to 31 December 2019	A prevalence of KC of 238 (5.5%) was observed among 4342 DS patients.	[13]
Keratoconus detection by novel indices in patients with Down syndrome: a cohort population-based study	250 DS patients	17.0 ± 4.7 years Range: 10–30 years	Iranians	Slit-lamp examinations, topographic indices were measured by Pentacam HR and corneal biomechanics were examined using the Corvis ST	A prevalence of KC of 28 (12.39%) was observed among the DS patients.	[71]
Corneal grafting for keratoconus in mentally retarded patients	33 DS patients	36.7 ± 10.8 years Range: 18–60 years	Norwegians	Not reported	KC was observed in 33 mentally retarded patients with DS.	[72]

Note: DS—Down syndrome, KC—keratoconus, MDS—Mothers with DS children, MNC—Mothers with normal children, OR—odds ratio, CCT—central corneal thickness, TP—Thinnest point of cornea, and CV—Corneal volume.

**Table 2 ijms-23-10796-t002:** Eye rubbing as a potential risk factor of KC in DS patients.

Title of Paper	Sample Size	Ethnicity	Extract from Study	Reference
Corneal Morphologic Characteristics in Patients with Down Syndrome	112 DS patients	DS group—59% Caucasians, and 41% Arabs	“Furthermore, it has been reported that patients with DS **frequently rub their eyes** which is a habit related to keratoconus development owing to the inflammation process and biomechanical alterations linked to eye rubbing habit”	[30]
Prevalence of ocular abnormalities in adults with Down syndrome in Hong Kong	91 DS patients	Chinese	“This, together with **frequent eye rubbing**, may predispose DS patients to keratoconus”	[43]
The Ocular Features of Down’s Syndrome	53 DS patients	Caucasians	“The cause of this increased incidence of keratoconus in Down’s syndrome is unknown. Several investigators have suggested **eye rubbing** as a probable cause”	[145]
Acute keratoconus with perforation in a patient with Down’s syndrome	1 DS patient	Austrian	“**Habitual eye rubbing**, which is frequently observed in patients with Down’s syndrome and other forms of mental deficiency, has been postulated as an important factor not only for the development of keratoconus itself”	[146]
Posterior corneal features in patients with Down syndrome and their relation with keratoconus	20 DS patients	Spanish	“Furthermore, patients with DS used to rub their eyes, and it is well known that the **eye rubbing** habit is an important risk factor for keratoconus development”	[36]
Computerized corneal topography in a pediatric population with Down syndrome	21 DS patients	Canadians	“Some authors have attributed the high prevalence of keratoconus in patients with Down syndrome to **chronic eye rubbing** because of irritation caused by blepharitis, seen in 7–47% of patients with Down syndrome”	[41]
Keratoconus and corneal morphology in patients with Down syndrome at a pediatric hospital	31 DS patients	16 Caucasians, 10 Hispanics, 3 “other” patients, and 1 unreported patient	“Eye rubbing may also play a role in development of keratoconus and **eye rubbing** is commonly reported in patients with DS”	[38]
Biometric measurements of the eyes in teenagers and young adults with Down syndrome	47 DS patients	Norwegians	“A thin cornea must be assumed to be particularly vulnerable to **eye rubbing**, and together these factors might lead to clinically manifest keratoconus in many Down syndrome patients”	[40]
Ocular findings in Malaysian children with Down syndrome	60 DS patients	Malaysians	“However, there has been no established evidence to link genetic abnormality of Down syndrome to keratoconus. It is thought to be due to **eye rubbing** or underlying structural abnormalities of the cornea”	[59]
Topographic screening reveals keratoconus to be extremely common in Down syndrome	48 DS patients	New Zealanders	“**Eye rubbing** behaviour is also common in DS and may further contribute to the predisposition to keratoconus development”	[147]
Acute Corneal Hydrops in Down Syndrome	1 DS patient	Japanese	“**Habitual eye rubbing**, which is frequently observed in patients with Down syndrome and other forms of mental deficiency, has been postulated as an important factor, not only for the development of keratoconus”	[148]
Evaluation of early corneal topographic changes in children with Down syndrome	27 DS patients	Caucasians	“**Habitual eye rubbing**, which is also frequently observed in patients with DS, has been postulated as a crucial factor either for the development of keratoconus or the progression of the disease”	[149]
Keratoconus detection by novel indices in patients with Down syndrome: a cohort population-based study	250 DS patients	Iranians	“**Eye rubbing** was reported in 18.1% of the healthy individuals, in 13.6% of those with suspected KC, in 19.6% of those with KC, and in 21.1% of those with progressive KC”	[71]
Severe acute corneal hydrops in a patient with Down syndrome and persistent eye rubbing	1 DS patient	Turkish	“**Eye-rubbing** may play a role in the pathogenesis of acute hydrops,”	[150]
Amniotic membrane transplantation with cauterization for keratoconus complicated by persistent hydrops in mentally retarded patients	8 DS patients	Polish	“**Vigorous eye rubbing** is a cause of extensive hydrops in mentally retarded patients with keratoconus”	[151]

Note: DS: Down syndrome, KC: keratoconus.

**Table 3 ijms-23-10796-t003:** Mouse models in DS.

Mouse Model	Description	Clinical Phenotype in DS	Limitation(s)	Reference(s)
Ts16 (Trisomy 16)	First mouse model released in 1980, trisomic for all parts of mouse chromosome 16 in synteny with human chromosome 21	Early developmental abnormalities	Barely survived to term by dying in utero, syntenic with other regions of human chromosome 3 and human chromosome 8	[181,182,183,184]
Ts65Dn	Segmentally trisomic for about 55% of genes on mouse chromosome 16 that are homologous to segment MRP139-ZNF295 on human chromosome 21	Impaired vision and hearing, congenital heart defects, learning and memory deficits	Difficulty in using model in vision research due to retinal degeneration, male mice are sterile	[181,183,185,186]
Ts1Cje	Produced from the reciprocal translocation between the distal portion of mouse chromosome 16 and the end of chromosome 12. Trisomic for a shorter region of mouse chromosome 16, which is smaller than that in Ts65Dn mouse spanning from *Sod1* to *Mx1* genes	Craniofacial abnormalities	Loss of a functional variant of superoxide dismutase1	[181,183]
Ts2Cje (Derivative of Ts65Dn)	Developed from Ts65Dn mice by a Robertsonian translocation of the extra chromosome of Ts65Dn mice onto mouse chromosome 12	Similar features as seen in Ts65Dn mice—congenital heart defects, learning and memory deficits	Although improved compared to the Ts65Dn, males still have moderate fertility	[181,187]
Tc1	First transchromosomic (Tc) mouse DS model with a nearly complete fragment of human chromosome 21 including most of the gene orthologs located on mouse chromosome 10, 16, and 17	Congenital heart defects, learning and memory deficits, motor coordination defects	Mosaic trisomy, where the human chromosome 21 is lost randomly in a large number of cells	[178,188,189]
TcMAC21	Humanized DS mouse model containing a clone of the long arm of the human chromosome 21 as an artificial mouse chromosome	Learning and memory deficits, craniofacial abnormalities, heart anomalies	Made up of some deletions that affect about 8% of genes found on the long arm human chromosome 21	[178,188]
Ts1Rhr	Segmental trisomic model containing segments of mouse chromosome 16 that are homologous to human chromosome 21. The model is also trisomic for the DS-critical region (DSCR) known to contain genes that may influence mental abnormalities in DS	Learning and memory impairment, craniofacial abnormalities	May not show accurate phenotypic feature in DS due to the inability of the trisomic region for the DSCR to impair hippocampal function in the brain	[190]

## Data Availability

Data sharing not applicable.

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
