# Peer review of "The Underlying Relationship between Keratoconus and Down Syndrome"

_ijms, 2022, doi:10.3390/ijms231810796_

Round 1

Reviewer 1 Report

Authors have meticulously written the present review about underlying relationship between Keratoconus and Down syndrome. The review provides comprehensive information on this important associative link between the two diseases that has been noted in various clinical studies over the years. The information is provided in a well structured manner and should be of use to the field of Keratoconus. There are a few minor comments as follows:

1)    Abstract, page1, line 28-29. Reconsider the statement “the available DS mouse models that are currently used in modeling KC in DS patients.” for better understanding to the reader. Have any DS mouse models actually been used for KC studies? As Table 3 for mouse model in DS has no reference of using either of the mice for modeling KC.

2)    Introduction, page1, line 46. Cite the relevant and recent refences for KC incidence worldwide.

3)    Page3, line 116-117. “PubMed literature search and compile all the studies of KC in DS, in both young and adult patients”. This statement is misleading, as selectively nineteen studies were included in table1. The authors are suggested to mention the selection criteria in the main text, if any.

4)    Page6, line 155. “Although in small proportions, collagen VI may have an integral function in the development of KC and DS.” Cite refence for this statement.

5)    Reference number 87 and 88 are the same. Authors are suggested to remove one and re-number the references throughout the manuscript.

6)    Page7, line 192. SOD1 in Down syndrome is reported in many papers. Kindly include more references for the same.

7)    Page7, line 205-206. “Several studies have reported that DS patients have significantly reduced corneal thickness at different ages.” References are missing after this statement.

8)    Summary, page 14. Authors are advised to include more corneal features in addition with CCT in KC and DS in section 4, page 7, to state “a strong connection between KC and DS in the area of corneal features”. Example: several corneal topographical studies are reported in DS.

9)    Schema 2, page 14. Kindly explain “Early detection and sensitivity of instrument used” meaning in context of relationship between KC and DS.

Author Response

Reviewer 1:

Authors have meticulously written the present review about underlying relationship between Keratoconus and Down syndrome. The review provides comprehensive information on this important associative link between the two diseases that has been noted in various clinical studies over the years. The information is provided in a well structured manner and should be of use to the field of Keratoconus. There are a few minor comments as follows:

Comment 1: Abstract, page1, line 28-29. Reconsider the statement “the available DS mouse models that are currently used in modeling KC in DS patients.” for better understanding to the reader. Have any DS mouse models actually been used for KC studies? As Table 3 for mouse model in DS has no reference of using either of the mice for modeling KC.

Response: Thank you for pointing this out. No DS models have been used in modeling KC. As suggested, we have rephrased the sentence on line 28-29 to “as well as the available DS mouse models that could be used in modeling KC in DS patients”.

Comment 2:  Introduction, page1, line 46. Cite the relevant and recent refences for KC incidence worldwide.

Response: Thank you for the note. As suggested, additional references (>5) have been included and cited in the manuscript for KC incidence worldwide in Introduction, page 1, line 46.

Comment 3:  Page3, line 116-117. “PubMed literature search and compile all the studies of KC in DS, in both young and adult patients”. This statement is misleading, as selectively nineteen studies were included in table1. The authors are suggested to mention the selection criteria in the main text, if any.

Response: Thank you for the advice. As suggested, selection criteria on nineteen studies have been added on page 3, line 116-120 in the main text.

Comment 4:  Page6, line 155. “Although in small proportions, collagen VI may have an integral function in the development of KC and DS.” Cite reference for this statement.

Response: Thanks for the suggestion. As suggested, additional references have been included to support this statement on page 6, line 157.

Comment 5:  Reference number 87 and 88 are the same. Authors are suggested to remove one and re-number the references throughout the manuscript.

Response: Thank you for pointing this out. As suggested, duplicated references have been removed. We also checked all the other references to avoid similar mistakes.

Comment 6:   Page7, line 192. SOD1 in Down syndrome is reported in many papers. Kindly include more references for the same.

Response: Thank you for the insight. As suggested, additional relevant references (>5) have been included on page 7, line 193.

Comment 7: Page7, line 205-206. “Several studies have reported that DS patients have significantly reduced corneal thickness at different ages.” References are missing after this statement.

Response: Thank you for the advice. As suggested, additional references have been included on page 7, line 208.

Comment 8: Summary, page 14. Authors are advised to include more corneal features in addition with CCT in KC and DS in section 4, page 7, to state “a strong connection between KC and DS in the area of corneal features”. Example: several corneal topographical studies are reported in DS.

Response: Thank you for the suggestion. As suggested, sentences of other corneal features have been included with relevant references on page 8, lines 228-234 and in summary, page 14, lines 346-347.

Comment 9: Schema 2, page 14. Kindly explain “Early detection and sensitivity of instrument used” meaning in context of relationship between KC and DS.

Response: Thank you for the advice. As suggested, “Early detection and sensitivity of instrument used” has been explained in the figure legend on page 15, lines 368-369.

Reviewer 2 Report

The authors made a comprehensive review on the relationship between KC and DS, which is well-written and can help us better understand both the diseases. I only have some minor suggestions:

1.       I suggest to add a paragraph on the method of literature search.

2.       There are a few recently published references, such as PMID 35367480 and 35795721. The authors can check whether they are related to the topic of the manuscript.

Author Response

Reviewer 2:

The authors made a comprehensive review on the relationship between KC and DS, which is well-written and can help us better understand both the diseases. I only have some minor suggestions:

Comment 1:    I suggest to add a paragraph on the method of literature search.

Response: Thank you for the suggestion. As suggested, selection criteria for articles in tables 1-3 have been described. Table 1 has been described in main text on page 3, lines 116-120. Table 2 has been described in main text on page 8, lines 256-257 and table 3 has been described in main text on page 12, lines 325-328.

Comment 2:  There are a few recently published references, such as PMID 35367480 and 35795721. The authors can check whether they are related to the topic of the manuscript.

Response: Thank you for this note. These two new reports (PMID 35367480 and 35795721) have been included on page 2, line 76 and on page 8 line 229.